# A Basic Study of Photodynamic Therapy with Glucose-Conjugated Chlorin e6 Using Mammary Carcinoma Xenografts

**DOI:** 10.3390/cancers11050636

**Published:** 2019-05-08

**Authors:** Tomohiro Osaki, Shota Hibino, Inoru Yokoe, Hiroaki Yamaguchi, Akihiro Nomoto, Shigenobu Yano, Yuji Mikata, Mamoru Tanaka, Hiromi Kataoka, Yoshiharu Okamoto

**Affiliations:** 1Joint Department of Veterinary Clinical Medicine, Faculty of Agriculture, Tottori University, Tottori 680-8553, Japan; mayrupi@yahoo.co.jp (S.H.); inoru.y@gmail.com (I.Y.); yokamoto@muses.tottori-u.ac.jp (Y.O.); 2Department of Applied Chemistry, Graduate School of Engineering, Osaka Prefecture University, 1-1 Gakuen-cho, Naka-ku, Sakai, Osaka 599-8531, Japan; hiroaki_yamaguchi@sanbo-chem.co.jp (H.Y.); nomoto@chem.osakafu-u.ac.jp (A.N.); 3KYOUSEI Science Center for Life and Nature, Nara Women’s University, Kitauoyahigashi-machi, Nara 630-8506, Japan; yano-s@cc.nara-wu.ac.jp (S.Y.); mikata@cc.nara-wu.ac.jp (Y.M.); 4Department of Gastroenterology and Metabolism, Nagoya City University Graduate School of Medical Sciences, 1 Kawasumi, Mizuho-cho, Mizuho-ku, Nagoya 467-8601, Japan; mtanaka77@gmail.com (M.T.); hkataoka@med.nagoya-cu.ac.jp (H.K.)

**Keywords:** photodynamic therapy, glucose-conjugated chlorin e6, canine mammary carcinoma

## Abstract

By using the Warburg effect—a phenomenon where tumors consume higher glucose levels than normal cells—on cancer cells to enhance the effect of photodynamic therapy (PDT), we developed a new photosensitizer, glucose-conjugated chlorin e6 (G-Ce6). We analyzed the efficacy of PDT with G-Ce6 against canine mammary carcinoma (CMC) in vitro and in vivo. The pharmacokinetics of G-Ce6 at 2, 5, and 20 mg/kg was examined in normal dogs, whereas its intracellular localization, concentration, and photodynamic effects were investigated in vitro using CMC cells (SNP cells). G-Ce6 (10 mg/kg) was administered in vivo at 5 min or 3 h before laser irradiation to SNP tumor-bearing murine models. The in vitro study revealed that G-Ce6 was mainly localized to the lysosomes. Cell viability decreased in a G-Ce6 concentration- and light intensity-dependent manner in the PDT group. Cell death induced by PDT with G-Ce6 was not inhibited by an apoptosis inhibitor. In the in vivo study, 5-min-interval PDT exhibited greater effects than 3-h-interval PDT. The mean maximum blood concentration and half-life of G-Ce6 (2 mg/kg) were 15.19 ± 4.44 μg/mL and 3.02 ± 0.58 h, respectively. Thus, 5-min-interval PDT with G-Ce6 was considered effective against CMC.

## 1. Introduction

Breast cancer is currently the most commonly diagnosed cancer and one of the deadliest diseases in women [1]. Conventional treatments for breast cancer include surgery, radiation therapy, chemotherapy, and hormone therapy. Photodynamic therapy (PDT) is a relatively new approach in treating cancer. PDT, a minimally invasive cancer treatment, relies on the combination of a photosensitizer, light, and oxygen to eliminate tumor cells or microorganisms. Although these are nontoxic individually, when combined, they initiate a photochemical reaction that culminates in the generation of a highly reactive species termed singlet oxygen (^1^O_2_) or radicals [2,3]. These reactive species can rapidly cause substantial toxicity, subsequently leading to cell death via apoptosis, necrosis, or autophagy [2,3]. Photosensitizers are selectively localized in tumors owing to the physiological differences between tumor and normal tissues [4,5]. Selectivity is derived from both the ability of photosensitizers to localize in neoplastic lesions and the precise delivery of light to the treated sites [2]. PDT is, therefore, safe and effective when used to selectively eradicate malignant and abnormal tissues and avoids systemic toxicity and side effects [2].

Porfimer sodium (Photofrin^®^) is a first-generation photosensitizer that weakly absorbs light at 630 nm [4]. Photofrin^®^ has been clinically accepted in several countries to treat esophageal cancer [6,7], lung cancer [8], and Barrett’s esophagus [7]. Treatment with porfimer sodium extends to Kaposi’s sarcoma, and head and neck, brain, intestinal, skin, breast, cervix, urinary bladder, abdominal, and thoracic cancers [9,10]. Porfimer sodium requires a long clearance time of 4–8 weeks after injection to avoid skin photosensitization [11]. Owing to properties such as unfavorable skin photosensitivity and low absorption in the red region of the visible spectrum, second-generation photosensitizers have been developed. Talaporfin sodium (NPe6) is a second-generation photosensitizer and hydrophilic chlorine agent derived from chlorophyll, with strong absorption at 664 nm [4,12,13]. NPe6 has also been approved in Japan under the tradename Laserphyrin (Meiji Seika Pharma, Tokyo, Japan) to treat malignant tumors such as lung cancer [14], brain tumor [15,16], esophageal cancer [17,18], and cholangiocarcinoma [19]. NPe6 has a short clearance time (only 3–7 days after injection) and causes minimal skin photosensitivity, unlike Photofrin^®^ [4,11]. Second-generation photosensitizers, however, exhibit poor body clearance and fail to exert sufficient tumor selectivity [20]. Therefore, the development of more effective photosensitizers is warranted.

Cancer cells consume more glucose via glycolytic fermentation for lactate formation than by respiration, a characteristic known as the Warburg effect [21]. Several glycol-conjugated porphyrins and chlorins have been synthesized and evaluated to determine their photodynamic effect [22,23,24,25,26]. However, these photosensitizers had drawbacks owing to their low solubility and low excretion rate from the body. Recently, Nishie et al. reported the therapeutic effect of PDT with the newly developed glucose-conjugated chlorin e6 (G-Ce6; Figure 1), which had improved water solubility and rapid excretion from the body [27]. Nevertheless, this report was based on murine models, which have been reported to have limitations [28]. Therefore, it was considered that cancer therapy should be studied in large animal models that more closely reflect the clinical settings.

Comparative oncology has been reported to be a quickly expanding field, which examines both cancer risk and tumor development across species [29]. Canine mammary carcinomas (CMCs) occur spontaneously and have similar clinical presentation and pathophysiology to human cancers [29,30]. In particular, canine invasive mammary carcinomas have a short disease course; therefore, dogs are potent, valuable, and spontaneous organisms for use as cancer models to test novel therapies [30,31]. Therefore, it was considered that studying dogs with naturally occurring cancer may provide a valuable perspective, which is different from that obtained when studying human or rodent cancers alone [32].

In the present study, for translation research, we analyzed the cell death mechanism of PDT with G-Ce6 in CMC cells in vitro. In addition, we evaluated the efficacy of PDT with G-Ce6 performed at 5-min and 3-h intervals to treat mice bearing CMCs. Moreover, we evaluated the pharmacokinetics of G-Ce6 in normal dogs.

## 2. Results

### 2.1. Subcellular Localization of G-Ce6

Fluorescence micrographs of SNP cells double-stained with the photosensitizer and the mitochondrial or lysosome probe are shown in Figure 2 and Figure 3, respectively. The overlaid images had yellow-green fluorescent spots, indicating an overlap of G-Ce6 (red) and the mitochondrial or lysosome probe (green). Fluorescence of G-Ce6 was observed to localize in the lysosome of SNP cells.

### 2.2. G-Ce6 and Light Induce Cytotoxicity in SNP Cells

Figure 4 shows the viability of SNP cells at 24 h after PDT with mono-L-aspartyl chlorin e6 (NPe6) (a) and G-Ce6 (b). Values of the half-maximal inhibitory concentration (IC_50_) of NPe6 against SNP cells exposed to light doses of 1, 5, and 15 J/cm^2^ were 75.2, 30.4, and 30.6 μg/mL, respectively. For G-Ce6, the IC_50_ values against SNP cells exposed to the above light doses were 33.4, 10.4, and 1.7 μg/mL, respectively. SNP cell survival was dependent on the concentration of G-Ce6, with significantly lower survival rates at higher G-Ce6 concentrations. Cell death was not observed in cells treated with <40 μg/mL G-Ce6 in the absence of laser irradiation. With 1.6 μg/mL G-Ce6, cell viability at 15 J/cm^2^ of light was significantly lower than that at 0 J/cm^2^ of light (*p* < 0.01). With 8 μg/mL G-Ce6, cell viability at 5 J/cm^2^ and 15 J/cm^2^ of light was significantly lower than that at 0 J/cm^2^ of light (*p* < 0.01). With 40 μg/mL G-Ce6, cell viability at 1, 5, and 15 J/cm^2^ of light was significantly lower than that at 0 J/cm^2^ of light (*p* < 0.01).

### 2.3. Apoptotic versus Necrotic Cell Death

Figure 5 shows SNP cells stained with Hoechst 33342 at 4 h after PDT. Hoechst 33342 was used to evaluate apoptosis because it allows for the visualization of nuclear condensation and fragmentation that are characteristic of apoptosis. Signs of cell death were not visible in the control and 0 J/cm^2^ PDT groups. In the 1 J/cm^2^ PDT group, however, cells exhibited membrane blebbing. In the 5 J/cm^2^ PDT group, cells showed shrinkage, whereas in the 15 J/cm^2^ PDT groups, cell membranes were not observed and a decrease in the number of SNP cells was evident. In the 5 and 15 J/cm^2^ PDT groups, nuclear condensation of cells was indicated by Hoechst 33342 staining.

Figure 6 shows the images of SNP cells stained using Annexin V-FITC and EthD-III at 4 h after PDT. In the 0 and 1 J/cm^2^ PDT groups, a few cells were positively stained with EthD-III, whereas in the 5 and 15 J/cm^2^ PDT groups, many cells were positively stained with Annexin V or EthD-III. This indicated the occurrence of phosphatidylserine translocation and the loss of plasma membrane integrity, implying that the cells were either in late apoptotic or in early necrotic stages.

### 2.4. Kinetics Experiment to Assess Apoptosis

Apoptosis induced by G-Ce6-mediated-PDT was determined using Annexin V-FITC fluorescence in a live-cell analysis system. Apoptosis was found to be dependent on the concentration of G-Ce6 (Figure 7). This was because at low G-Ce6 concentration (<0.064 μg/mL), the total green object integrated intensity (green calibrated unit (GCU) × μm²/image) did not increase after PDT, whereas at 0.32 μg/mL G-Ce6 concentration, GCU began to increase at 1–3 h after PDT and reached a maximum at 10 h after PDT. Treatment with 8 μg/mL G-Ce6 resulted in apoptosis immediately after PDT and reached a plateau at approximately 4 h. Therefore, we assessed the cells at 4 h after PDT in the following studies. These results indicated that the GCU level is dependent on the concentration of G-Ce6.

### 2.5. Analysis of Apoptosis

At 4 h after PDT with a laser energy of 1, 5, or 15 J/cm^2^, the apoptotic rates in the PDT (treated with 8 μg/mL G-Ce6 and then irradiated with a light dose of 1, 5, or 15 J/cm^2^) groups were significantly higher than those in the control group (all *p* < 0.01; Figure 8). Furthermore, the percentage of apoptotic cells following PDT increased in a light-dependent manner.

### 2.6. Analysis of Reactive Oxygen Species

The percentage of reactive oxygen species (ROS) (+) cells at 4 h after PDT is shown in Figure 9. The percentage of ROS (+) cells in the 5 J/cm^2^ PDT group was significantly higher than that in the control group (*p* < 0.01). As cell membranes were not observed in the 15 J/cm^2^ PDT groups, the percentage of ROS (+) cells remarkably decreased.

### 2.7. Effect of a Pan-Caspase Inhibitor on G-Ce6-PDT-Induced Cell Death

SNP cells were treated with G-Ce6-PDT in the presence (10 μM) or absence of Z-VAD-FMK. G-Ce6-PDT-induced apoptosis and cell death could not be blocked by Z-VAD-FMK (Figure 10).

### 2.8. Intratumoral Localization

Figure 11 shows the intratumoral localization of G-Ce6 in SNP tumors 5 min and 3 h after administration (10 mg/kg, intravenously (i.v.)). At 5 min after administration, G-Ce6 fluorescence was observed within the tumor blood vessel; conversely, at 3 h post administration, fluorescence was apparent in both tumor interstitial tissue and tumor cells.

### 2.9. Tumor Response to PDT

Unlike in the control group, PDT induced significant tumor regrowth delay in the 5-min-interval PDT as well as in the 3-h-interval PDT groups (*p* < 0.05; Figure 12). In addition, the 5-min-interval PDT was more effective in inducing tumor regrowth delay than the 3-h-interval PDT. The mean tumor weight on day 25 was 210.3 ± 85.0 mg and 34.0 ± 12.7 mg in the 3-h-interval PDT and 5-min-interval PDT groups, respectively (Figure 13).

### 2.10. Histological Examination

A histological examination of untreated tumors showed tumor cells infiltrating the dermis, intact blood vessels at the tumor periphery, and necrotic tumor cells in the center region of tumors (Figure 14a,d).

At 24 h after the 5-min-interval PDT, widespread tumor cells appeared necrotic, and fibrin thrombus formation within the vessels was observed in the surrounding normal subcutaneous tissues (Figure 14b,e). At 24 h after the 3-h-interval PDT, cells at the superficial tumor tissue appeared necrotic, whereas deep-seated tumor cells appeared intact (Figure 14c,f).

### 2.11. Pharmacokinetics

The mean (± standard deviation (S.D.)) plasma G-Ce6 concentration–time curve obtained after intravenous injection fitted well to a two-compartment model (Figure 15). The data indicated a rapid initial distribution of G-Ce6 within the central compartment, followed by an apparently slower elimination phase. 

In the 2 mg/kg group (*n* = 6), mean C_max_ = 15.19 ± 4.44 μg/mL; mean T_1/2_ = 3.02 ± 0.58 h; mean area under the curve (AUC) = 12.03 ± 3.18 μg/mL·h; mean Cl = 175.9 ± 45.68 mL/h∙kg; and V_dss_ = 529.19 ± 188.67 mL/kg. In the 5 mg/kg group (*n* = 6), mean C_max_ = 46.99 ± 12.22 μg/mL; mean T_1/2_ = 2.84 ± 0.60 h; mean AUC = 59.66 ± 7.87 μg/mL·h; mean Cl = 84.75 ± 10.73 mL/h∙kg; and V_dss_ = 258.11 ± 65.57 mL/kg. In the 20 mg/kg group (*n* = 1), C_max_ = 82.05 μg/mL; T_1/2_ = 2.51 h; AUC = 234.75 μg/mL·h; Cl = 85.2 mL/h∙kg; and V_dss_ = 284.40 mL/kg (Table 1).

### 2.12. General Physical Examination and Blood Analysis

The dose of 2 mg/kg G-Ce6 was well tolerated by dogs in the safety examination study conducted. There was no evidence of major skin photosensitivity or acute toxicity such as vomiting, diarrhea, or salivation. No significant changes in serum biochemical parameters were observed following the administration of G-Ce6 (Table 2). Nausea was observed during the administration of 5 mg/kg G-Ce6; however, no significant changes in serum biochemical parameters were observed after administering G-Ce6. Side effects such as vomiting and weakness were observed during the administration of 20 mg/kg G-Ce6. Increases in the levels of aspartate transaminase (AST) and alanine transaminase (ALT) in blood was also observed. Based on the results accumulated, 20 mg/kg G-Ce6 was considered the toxic dose level.

Glucose (Glu), aspartate transaminase (AST), alanine transaminase (ALT), alkaline phosphatase (ALP), γ-glutamyltranspeptidase (GGT), blood urea nitrogen (BUN), and creatinine (Cre); 2 mg/kg (*n* = 6), 5 mg/kg (*n* = 3), and 20 mg/kg (*n* = 1). The results are presented as the mean ± standard deviation.

## 3. Discussion

We developed a new PDT approach with G-Ce6, which showed improved water solubility and rapid excretion from the body [27]. However, a solvent, such as Tween 80, was required to resolve G-Ce6. Upregulation of glucose transporters (GLUTs) has been reported in numerous cancer types [33] as they enable the sustenance of the energy demand required by tumor cells for various biochemical programs. GLUTs may, therefore, serve as promising anticancer targets. It has been reported that three glucose transporters (GLUT1, GLUT3, and GLUT4) might play crucial roles in the cellular uptake of glucose-conjugated chlorin (5,10,15,20-tetrakis[4-[b-D-glucopyranosylthio-2,3,5,6-tetrafluorophenyl]2,3,[methano[N-methyl] iminomethano] chlorin) into GIST-T1 cells [24]. In addition, PDT with G-Ce6 has been reported to show very strong antitumor effects in murine colorectal cancer-bearing mice [27].

In the in vitro study, G-Ce6 was observed to primarily accumulate in the lysosomes. The accumulation of photosensitizers in cell organelles is dependent on the charge of the sensitizer. Cationic compounds accumulate in the mitochondria, whereas anionic compounds accumulate in the lysosomes [4]. As G-Ce6 is uncharged, the accumulation of G-Ce6 in the cell might not be related to the charge of the photosensitizer.

The cells were incubated with various concentrations of G-Ce6 for 1 h. After washing with fresh media, the cells were irradiated with 650-nm laser light (10 mW/cm^2^; 0, 1, 5, and 10 J/cm^2^). No cytotoxicity was observed at 1 J/cm^2^ (data not shown). To evaluate the IC_50_ values at a light dose of 1 J/cm^2^, the cells were incubated with various concentrations of G-Ce6 for 24 h. G-Ce6-mediated PDT-induced cell death in SNP cells was dependent on the photosensitizer dose and the light dose. Dark cytotoxicity of G-Ce6 was not observed. We also evaluated the cytotoxicity of PDT with G-Ce6 compared to NPe6, a hydrophilic chlorin agent derived from chlorophyll. The IC_50_ values of NPe6 against SNP cells were 75.2, 30.4, and 30.6 μg/mL at light doses of 1, 5, and 15 J/cm^2^, respectively. The IC_50_ values of G-Ce6 against SNP cells were 33.4, 10.4, and 1.7 μg/mL at light doses of 1, 5, and 15 J/cm^2^, respectively. G-Ce6-mediated PDT at 15 J/cm^2^ induced a 30-fold greater cytotoxicity than NPe6. It has been reported that the IC_50_ value of G-Ce6 against murine colon cancer cells was 1.35 nM at a light dose of 16 J/cm^2^ (1.2 μg/mL) [27]. Thus, the in vitro antitumor effects of PDT with G-Ce6 was superior to that with NPe6. G-Ce6-mediated PDT also increased the percentage of Annexin V- and ROS-positive SNP cells in a light dose-dependent manner. However, the percentage of ROS-positive cells at 15 J/cm^2^ was lower than that at 5 J/cm^2^. Cells treated at 15 J/cm^2^ were severely destroyed; therefore, we were unable to accurately measure the number of ROS-positive cells. Cells treated with 10 μg/mL G-Ce6 and a light dose of 15 J/cm^2^ underwent either late apoptosis or early-stage necrosis (Figure 6). Z-VAD-FMK is a specific general inhibitor of caspase; nevertheless, this inhibitor did not influence the number of Annexin V-positive cells and the cytotoxicity induced by G-Ce6-mediated PDT. The mechanism underlying cell death induced by G-Ce6-mediated PDT might, therefore, be necrosis induced by ROS production.

The biological target of PDT is dependent on the interval between the injection of drug (photosensitizer) and the light irradiation. PDT with a short drug–light interval targets tumor vasculature (vascular-targeting PDT), whereas PDT with a long drug–light interval targets cellular compartments (cellular-targeting PDT) [34]. We used vascular-targeting PDT (5-min-interval PDT) when G-Ce6 was primarily localized to the tumor vasculature (Figure 11a) or used cellular-targeting PDT (3-h-interval PDT) when G-Ce6 was primarily localized to the tumor cells (Figure 11b) in an in vivo study. Both PDT treatments inhibited tumor growth, with the 5-min-interval PDT being more effective than the 3-h-interval PDT. To confirm the effects of G-Ce6-mediated PDT, we performed histological examination of the tumor tissues. At 24 h following the 3-h-interval PDT, cells in the superficial tumor tissue appeared necrotic; however, deep-seated tumor cells appeared intact. The treatments were initiated when the average tumor volume reached about 100 mm^3^ (height: about 4 mm) in the tumor model. Penetration of light is 3–8 mm for a wavelength of 630–800 nm [35]. Hence, light could sufficiently penetrate the entire tumor in this study. Although the concentration of G-Ce6 at the surface and bottom of tumor was not quantified, G-Ce6 was considered to be heterogeneously distributed within tumors. Therefore, the efficacy of the 3-h-interval PDT varied based on the different parts of the tumor.

In contrast, widespread tumor cells appeared necrotic at 24 h after the 5-min-interval PDT. Vascular-targeting PDT has been reported to induce blood flow impairment and blood vessel destruction, resulting in tissue hypoxia, nutrient deprivation, and tumor destruction [34,36,37]. It was also reported that vascular-targeting PDT using benzoporphyrin derivative monoacid ring A could induce long-term tumor regression [37,38]. Therefore, the 5-min-interval PDT using G-Ce6 more effectively induced tumor growth delay than the 3-h-interval PDT with G-Ce6. Further studies are required to assess the effects of PDT with G-Ce6 against canine endothelial cells.

We analyzed the pharmacokinetics of G-Ce6 using healthy dogs as animal models for translational research. Based on the results of the pharmacokinetics and safety examination, 2 mg/kg G-Ce6 was identified as the optimal dose, which did not cause any adverse effects. After intravenous administration of G-Ce6 at a dose of 2 mg/kg to healthy beagle dogs, drug distribution occurred in the first hour, followed by an elimination phase with a mean *T*_1/2_ of 3.02 ± 0.58 h; the AUC of G-Ce6 at a dose of 2 mg/kg was 12.03 ± 3.18 μg/mL·h. In contrast, the *T*_1/2_ and AUC of porfimer sodium were 44.22 ± 8.10 h and 1670.2 ± 71.6 μg/mL∙h, respectively [39]. Intravenously administered G-Ce6 was rapidly cleared from the plasma unlike porfimer sodium; therefore, laser light had to be irradiated within a short time after photosensitizer administration to induce vascular-targeting PDT effects. Moreover, it was considered that the light shielding duration of G-Ce6 could be a few days.

Nishie et al. reported the therapeutic effect of PDT with G-Ce6 using a murine model [27]. However, to our knowledge, the present study is the first using G-Ce6 for CMC and pharmacokinetic studies of G-Ce6 using healthy dogs. PDT using G-Ce6 was effective for CMCs and G-Ce6 was rapidly cleared from the plasma. These results suggested that CMCs are useful for preclinical research in comparative oncology and would be an optimal design for human clinical trials. 

## 4. Materials and Methods

### 4.1. Cell Culture

The SNP CMC cells were established by the author [40]. SNP cells were cultured in 250 mL tissue culture flask (Corning Incorporated, Corning, NY, USA) containing RPMI 1640 medium (Invitrogen; Thermo Fisher Scientific, Inc., Waltham, MA, USA) supplemented with 10% heat-inactivated fetal bovine serum (Nichirei Biosciences, Tokyo, Japan) and PSN (5 mg/mL penicillin, 5 mg/mL streptomycin, and 10 mg/mL neomycin) solution (Invitrogen), then incubated in 5% CO_2_ at 37 °C. The cells were washed with phosphate-buffered saline for subculturing, then harvested from near-confluent cultures via a brief exposure to a solution containing 0.25% trypsin and 1 mmol/L tetrasodium ethylenediaminetetraacetic acid with phenol red (Invitrogen). Trypsinization was terminated using RPMI 1640 medium containing 10% fetal bovine serum. Trypsinized cells were transferred to a new tissue culture flask. 

### 4.2. Photosensitizer

Methyl (7S,8S)-18-ethyl-5-(2-methoxy-2-oxoethyl)-7-(3-methoxy-3-oxopropyl)-2,8,12,17-tetramethyl-13-(1-(3-(((2S,3R,4S,5S,6R)-3,4,5-trihydroxy-6-(hydroxymethyl) tetrahydro-2H-pyran-2-yl) thio) propoxy) ethyl)-7H,8H-porphyrin-3-carboxylate (G-Ce6, MW: 893.06) was synthesized and provided by the laboratory at Osaka Prefecture University (Osaka, Japan). For the in vitro and in vivo studies, G-Ce6 was dissolved in 20% Tween 80 (Polyoxyethylene(20) Sorbitan Monooleate, Wako Pure Chemical Co., Osaka, Japan) in phosphate buffered saline (Figure 1b,c). For intravenous administration to healthy dogs, G-Ce6 was dissolved in 2.5% Kolliphor (Sigma-Aldrich, St. Louis, MO, USA) and 97.5% saline. Figure 1d shows a UV-vis spectrum for G-Ce6 in DMSO (3 × 10^−6^ M). Figure 1e shows a HPLC spectum for G-Ce6. NPe6 was obtained from Meiji Seika Kaisha (Tokyo, Japan).

### 4.3. Subcellular Localization of G-Ce6

We cultured 1 × 10^5^ cells in a 35-mm Petri dish (Thermo Fisher Scientific, Waltham, MA, USA). Cells were then incubated with G-Ce6 at a final concentration of 10 μg/mL in complete cell culture medium for 6 h, followed by co-incubation with 50 nM MitoTracker Green FM (Invitrogen) and 50 nM LysoTracker Yellow HCK-123 (Invitrogen) for an additional 30 min at 23 °C in the culture medium before fluorescence microscopy. The fluorescence of G-Ce6 was detected with a filter (excitation, 405 nm; emission, 640 nm) using an all-in-one fluorescence microscope (BZ-X800, Keyence Co, Osaka, Japan). A BZ-X filter GFP (excitation, 470 nm; emission, 525 nm) was used to observe the mitochondria and lysosome; green and red images were combined to form an overlay image.

### 4.4. Evaluation of the Cytotoxic Effects of G-Ce6 and Light in SNP Cells

We seeded 1 × 10^4^ SNP cells into each well of 96-well plates (Corning Inc., New York, NY, USA) followed by overnight incubation. Cells were then incubated with various concentrations of G-Ce6 for 24 h at 37 °C. After washing with fresh media, the cells were irradiated with 650-nm light emitted by a semiconductor laser (Osada Electric Co., Ltd., Tokyo, Japan), using an optical fiber with a micro-lens delivery attachment (Pioneer Optics, Inc., Windsor Lock, CT, USA). PDT was performed at 10 mW/cm^2^ in cells exposed to six concentrations of NPe6 and G-Ce6 (0, 0.064, 0.32, 1.6, 8.0, or 40.0 μg/mL) using four light doses (0, 1, 5, or 15 J/cm^2^). The cells were then incubated for 24 h in the dark prior to examining cell viability using the Cell Counting Kit-8 (Dojindo, Kumamoto, Japan), according to the manufacturer’s instructions.

### 4.5. Analysis of Cell Death

We seeded 5 × 10^4^ SNP cells in 35-mm Petri dishes containing 2 mL of cultivation medium. After 24 h of incubation, the dishes were divided into the following four groups: 1) control group (no treatment); 2) 0 J/cm^2^ PDT group (treated with 8 μg/mL G-Ce6 and irradiated with light dose of 0 J/cm^2^); 3) 1 J/cm^2^ PDT group (treated with 8 μg/mL G-Ce6 and irradiated with light dose of 1 J/cm^2^); 4) 5 J/cm^2^ PDT group (treated with 8 μg/mL G-Ce6 and irradiated with light dose of 5 J/cm^2^); and 5) 15 J/cm^2^ PDT group (treated with 8 μg/mL G-Ce6 and irradiated with light dose of 15 J/cm^2^).

The cells were incubated with 8 μg/mL G-Ce6 for 24 h. After washing with fresh media, the cells were irradiated with 650-nm laser light (10 mW/cm^2^; 1, 5, and 15 J/cm^2^) emitted by a semiconductor laser using an optical fiber with a microlens delivery attachment. After 4 h of PDT, the cells were stained using the Promokine Apoptotic/Necrotic/Healthy cell detection kit (PromoKine, Heidelberg, Germany), according to the manufacturer’s instructions. The cells were then stained with Hoechst 33342 at 24 h after laser irradiation. Nuclear morphology was examined using an Olympus BX51 (Olympus Co., Tokyo, Japan). To assess apoptosis and necrosis, the cells were stained with Annexin V-fluorescein isothiocyanate (FITC) and ethidium homodimer III (EthD-III) at 24 h after laser irradiation. Nuclear morphology was examined using an Olympus Fluoview FV1000 (Olympus Co., Tokyo, Japan). The cells were analyzed using fluorescence microscopy using FITC and Texas Red filter setting.

### 4.6. Kinetics Experiment to Assess Apoptosis

SNP cells were seeded at a density of (1–2) × 10^4^ cells/well in 96-well plates and incubated for 24 h. The cells were then incubated with various concentrations of G-Ce6 (0, 0.064, 0.32, 1.6, or 8.0 μg/mL) for 24 h at 37 °C. After washing with fresh media, the cells were irradiated with 650-nm laser light (10 mW/cm^2^, 5 J/cm^2^) emitted by a semiconductor laser, using an optical fiber with a microlens delivery attachment. The cells were then stained with IncuCyte^®^ Annexin V (Essen BioScience Inc., Ann Arbor, MI, USA) immediately following PDT, and incubated in a 5% CO_2_ atmosphere at 37 °C for 24 h. Images were automatically captured every 1 h for 24 h in phase-contrast and fluorescence modes using the IncuCyte^®^ S3 system; phase contrast and green-phase images were obtained from the system. The total green object integrated intensity (green calibrated unit (GCU) × μm²/image) in an image was determined by object counting with IncuCyte^®^ S3 Software according to the manufacturer’s protocol. In the present study, object counting is defined as the total sum of the intensity of Annexin V Green fluorescence in the image.

### 4.7. Analysis of Apoptosis and Reactive Oxygen Species

SNP cells were seeded at a density of 5 × 10^4^ cells/well in 35-mm petri dishes containing 2 mL culture medium. Following 24 h of incubation, the cells were divided into the following groups: control (no treatment); laser (irradiated with a light dose of 15 J/cm^2^); G-Ce6 (treated with 8 μg/mL G-Ce6); and PDT (treated with 8 μg/mL G-Ce6, and then irradiated with a light dose of 1, 5, or 15 J/cm^2^). 

The cells were incubated with 8 μg/mL G-Ce6 for 24 h. After washing with fresh media, the cells were irradiated with 650-nm laser light (10 mW/cm^2^; 1, 5, and 15 J/cm^2^) emitted by a semiconductor laser, using an optical fiber with a microlens delivery attachment. Apoptosis was assessed 4 h after laser irradiation using the Muse Annexin V and Dead Cell Assay kit (EMD Millipore Co., Billerica, MA, USA), according to the manufacturer’s protocols. Annexin V was used to detect phosphatidylserine on the external membrane of apoptotic cells. 

Reactive oxygen species (ROS) generation was assessed 4 h after laser irradiation using the Muse Oxidative Stress kit (EMD Millipore Co.), according to the manufacturer’s protocols; this kit determines the percentage of cells that are negative (healthy cells) and positive for ROS (cells containing ROS).

Single-cell suspensions were then loaded onto the Muse Cell Analyzer (EMD Millipore Co.).

### 4.8. Effect of a Pan-Caspase Inhibitor on G-Ce6-PDT-Induced Cell Death

SNP cells were incubated with 8 μg/mL G-Ce6 for 24 h in the presence (10 μM) or absence of Z-VAD-FMK (Medical & Biological Laboratories Co. Ltd., Aichi, Japan), which is a pan-caspase inhibitor. After washing with fresh media, the cells were irradiated with 650-nm laser light (10 mW/cm^2^, 5 J/cm^2^) emitted by a semiconductor laser, using an optical fiber with a microlens delivery attachment. Apoptosis was assessed at 4 h after laser irradiation using the Muse Annexin V and Dead Cell Assay kit.

### 4.9. Tumor Models

Six-week-old female NOD/ShiJic-scidJcl mice (CLEA Japan, Tokyo, Japan) were maintained in isolators under specific pathogen-free conditions. All SCID mice were housed and handled under clean conditions using a Jic rack (Jic Co., Kanagawa, Japan), sterilized water, cages, beddings, and food (CL-2; CLEA Japan, Tokyo, Japan). Food and water were given ad libitum. SNP cells were inoculated subcutaneously in the shaved lower dorsum of mice at a density of 1 × 10^6^ cells in 0.1 mL phosphate buffered salts (Nacalai Tesque Inc., Kyoto, Japan) per mouse using a 26-gauge syringe. The treatments were initiated when the average tumor volume reached about 100 mm^3^ in tumor model. The use of animals and the procedures employed in this study were approved by the Animal Research Committee of Tottori University (project number: 15-T-48).

### 4.10. Intratumoral Localization

To determine the intratumoral localization of G-Ce6, tumors were excised either 5 min or 3 h after G-Ce6 (10 mg/kg, i.v.) administration. Samples were immediately mounted in Tissue-Tek^®^ O.C.T. compound (Sakura^®^ Finetek., Tokyo, Japan) and snap-frozen in dry ice with acetone. Cryostat cross sections (10-μm slices) were taken from the center of each tumor. Each section was examined using the all-in-one fluorescence microscope with a filter (excitation, 405 nm; emission, 640 nm).

### 4.11. Photodynamic Therapy of the Tumor

We examined the following two experimental groups: (i) a 3-h-interval PDT group (G-Ce6 administered 3 h before laser irradiation) and (ii) a 5-min-interval PDT group (G-Ce6 administered 5 min before laser irradiation). Tumor-bearing mice were intravenously administered approximately 0.1 mL of G-Ce6 stock solution to achieve a dose of 10 mg/kg body weight; they were irradiated with 677-nm laser light emitted by a diode laser (Osada Electric Co., Ltd., Tokyo, Japan). Light was delivered to the mice through a quartz fiber fitted with a microlens and expanded onto the tumor at a 2–4 mm skin margin. The tumors were exposed at a fluence rate of 250 mW/cm^2^, at a light dose of 100 J/cm^2^.

The resulting treatment effect was assayed by changes in tumor volume. The tumor size was regularly measured after PDT using a caliper. Tumor volume was calculated using the following formula: tumor volume (V) = (a × b × c) × π/6, where a, b, and c are three orthogonal diameters of the tumor. The experiment was terminated on day 25. All mice were sacrificed via cervical dislocation, and the tumors were harvested.

### 4.12. Histological Examination

To examine PDT-induced histological changes in the tumor tissue and surrounding normal tissue, animals were sacrificed via cervical dislocation, and the tumors were harvested at 24 h after PDT. The tumor tissues were fixed in 10% buffered formalin and embedded in paraffin. Tumor sections were cut at a thickness of 4 μm, stained with hematoxylin & eosin (H&E), and examined under a light microscope (Olympus BX51).

### 4.13. Analysis of Pharmacokinetics

The animal protocol using healthy beagle dogs was approved by the Animal Research Committee of the Tottori University (project number: 17-T-3). For the pharmacokinetic crossover study, 6, 3, and 1 healthy beagle dogs were intravenously administered G-Ce6 at a dose of 2, 5, and 20 mg/kg, respectively. The ages of the healthy dogs (3 males and 3 females) ranged from 6–12 years old (mean and median, 8.6 and 7.5 years, respectively); the dogs weighed 8.7–11.2 kg (mean and median, 10.2 and 10.5 kg, respectively).

The volume of G-Ce6 required to achieve the desired doses was diluted with 0.5% saline to obtain a total injection volume of 10 mL. The full injection volume was intravenously administered over 10 min at a rate of 60 mL/h using an appropriate syringe pump.

Blood samples were collected at 0, 10, and 30 min, and 1, 3, 6, and 24 h after the end of G-Ce6 injection. The samples were mixed with heparin in plastic tubes and centrifuged at 600 × *g* for 5 min at 4 °C; 0.2 mL of plasma from each centrifuged sample was transferred to Eppendorf tubes. The collected plasma samples were kept in the dark to avoid light exposure until assay initiation. For fluorescence measurement, plasma was read on a Fluorescence Spectrometer SH-9000Lab (excitation, 405 nm and emission, 665 nm; Hitachi High-Technologies Co., Tokyo, Japan). The concentration of G-Ce6 was determined by comparison with appropriate standards. The concentrations of G-Ce6 in the plasma were fitted to a two-compartment model and analyzed with a pharmacokinetic analysis program (http://www.pharm.kyoto-u.ac.jp/byoyaku/Kinetics/download.html). The pharmacokinetic parameters were calculated as follows: half-life (*T*_1/2_) = ln 2/β, where β is the slope of the elimination phase; volume of distribution at steady state (V_dss_) = dose (A/α^2^ + B/β^2^)/((A/α) + (B/β))^2^, where A and B are the intercepts of the distribution and elimination phases, respectively; α is the slope of the distribution phase; and clearance (CL) = dose/AUC, where AUC is the area under the curve, which was calculated using the trapezoidal method.

### 4.14. General Physical Examination and Blood Analysis

A general physical examination was performed on all dogs before and after PDT. Serum biochemistry analysis of glucose (Glu), AST, ALT, alkaline phosphatase (ALP), γ-glutamyltranspeptidase (GGT), blood urea nitrogen, and creatinine (Cre) was also performed for all dogs immediately before and 1 day after G-Ce6 administration.

### 4.15. Statistical Analysis

For the in vitro study, data were analyzed using Tukey’s multiple comparison test and Dunn’s multiple comparison test following two-way analysis of variance. For the in vivo study, data were analyzed using Bonferroni’s multiple comparison test following two-way analysis of variance. Statistical analyses were performed using GraphPad Prism version 6 (GraphPad Software Inc., La Jolla, CA, USA). A *p*-value < 0.05 indicated statistical significance. The results are presented as the mean ± S.D.

## 5. Conclusions

In summary, G-Ce6-mediated PDT-induced cell death in SNP cells was dependent on the photosensitizer dose and light dose. The 5-min-interval PDT using G-Ce6 may be beneficial in the treatment of dogs with mammary carcinoma and may shorten treatment time. However, there is a limitation associated with the in vitro cytotoxicity assay. In the present study, the incubation time of G-Ce6 in the in vitro cytotoxicity assay was 24 h. Considering the results of the in vivo study, the in vitro cytotoxicity assay needed to assess the photocytotoxicity of the SNP cells that were incubated with G-Ce6 for a short time. In the future, we will perform PDT with G-Ce6 in dogs with CMCs.

## Figures and Tables

**Figure 1 cancers-11-00636-f001:**
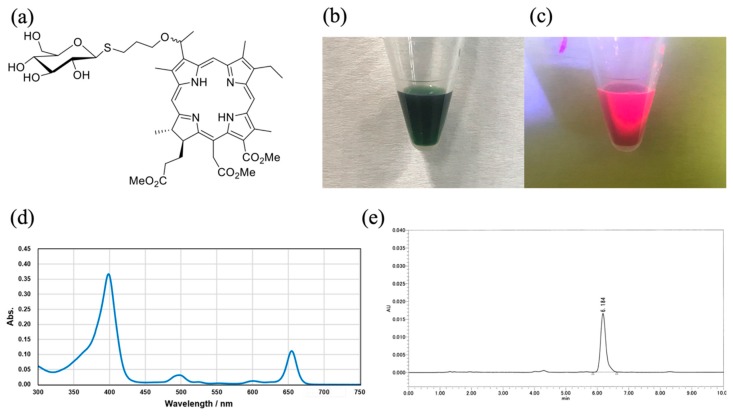
Glucose-conjugated chlorin e6 (G-Ce6). (**a**) Chemical structure of G-Ce6. Methyl (7S,8S)-18-ethyl-5-(2-methoxy-2-oxoethyl)-7-(3-methoxy-3-oxopropyl)-2,8,12,17-tetramethyl-13-(1-(3-(((2S,3R,4S,5S,6R)-3,4,5-trihydroxy-6-(hydroxymethyl) tetrahydro-2H-pyran-2-yl) thio) propoxy) ethyl)-7H,8H-porphyrin-3-carboxylate (G-Ce6). (**b**) White light image of 2 mg/mL G-Ce6 solution (20% Tween 80 in phosphate buffered saline). (**c**) Fluorescence image of 2 mg/mL G-Ce6 (excitation, 405 nm). (**d**) UV-vis spectrum for G-Ce6 in DMSO (3 × 10^−6^ M). (**e**) HPLC spectrum for G-Ce6. Column: Kinetex XB-C18 (2.6 μm (particle size), 4.6 mm (internal diameter), 75 mm (length), Phenomenex), 10 mmol/L ammonium acetate/acetonitrile (35:65, *v*/*v*) as solvent, and UV detector at 254 nm were used.

**Figure 2 cancers-11-00636-f002:**
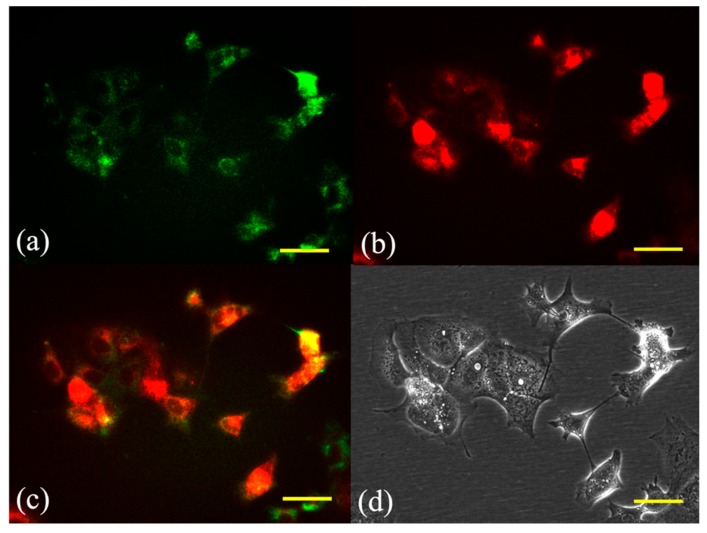
Subcellular localization of glucose-conjugated chlorin e6 in the mitochondria after 6 h of incubation. The subcellular localization of glucose-conjugated chlorin e6 (G-Ce6) was characterized using fluorescence microscopy. (**a**) Green fluorescence of MitoTracker Green FM stained mitochondria, (**b**) red fluorescence of G-Ce6 in the same view as (**a**), (**c**) combination of (**a**) and (**b**) images, and (**d**) transmission images. Scale bar, 50 μm.

**Figure 3 cancers-11-00636-f003:**
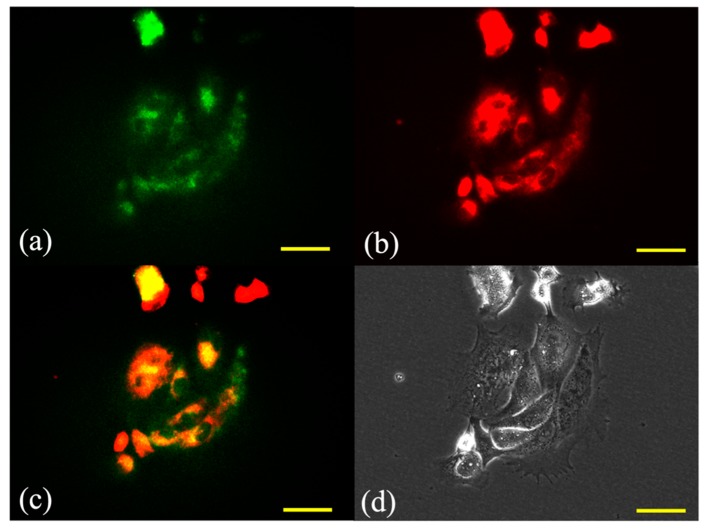
Subcellular localization of glucose-conjugated chlorin e6 in the lysosome after 6 h of incubation. The subcellular localization of glucose-conjugated chlorin e6 (G-Ce6) was characterized using fluorescence microscopy. (**a**) Green fluorescence of LysoTracker Yellow HCK-123 stained lysosome, (**b**) red fluorescence of G-Ce6 in the same view as (**a**), (**c**) combination of (**a**) and (**b**) images, and (**d**) transmission images. The images showed that G-Ce6 mainly localized in the lysosomes. Scale bar, 50 μm.

**Figure 4 cancers-11-00636-f004:**
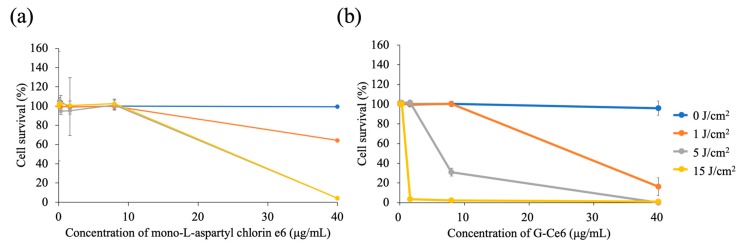
Photodynamic cytotoxicity in SNP cells. The cells were incubated with various concentrations of (**a**) mono-L-aspartyl chlorin e6 (NPe6) and (**b**) glucose-conjugated chlorin e6 (G-Ce6) for 24 h at 37 °C. After washing with fresh media, the cells were irradiated with 650-nm laser light (10 mW/cm^2^; 0, 1, 5, and 15 J/cm^2^). The viability of SNP cells at 24 h after photodynamic therapy, along with IC_50_ values, were determined. (**a**) The IC_50_ values of NPe6 at light doses of 1, 5, and 15 J/cm^2^ were 75.2, 30.4, and 30.6 μg/mL, respectively. (**b**) The IC_50_ values of G-Ce6 at light doses of 1, 5, and 15 J/cm^2^ were 33.4, 10.4, and 1.7 μg/mL, respectively. IC_50_, half-maximal inhibitory concentration. Results are presented as the mean ± standard deviation.

**Figure 5 cancers-11-00636-f005:**
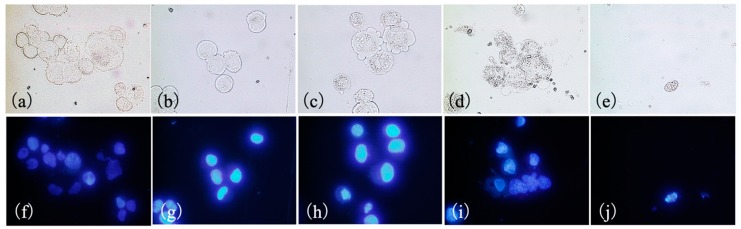
Morphological changes in SNP cells 4 h after photodynamic therapy (PDT) with glucose-conjugated chlorin e6 (G-Ce6). The cells were incubated with 10 μg/mL G-Ce6 for 24 h. After washing with fresh media, the cells were irradiated with 650-nm laser light (10 mW/cm^2^; 0, 1, 5, and 15 J/cm^2^). (**a,f**) Control, (**b,g**) 0 J/cm^2^, (**c,h**) 1 J/cm^2^, (**d,i**) 5 J/cm^2^, (**e,j**) 15 J/cm^2^. Upper panel: transmitted light images (a–e). Lower panel: fluorescence images (f–j). The cells were stained with Hoechst 33342 dye 4 h after laser irradiation. No signs of apoptosis were observed in the control and 0 J/cm^2^ PDT groups (**a**,**b**). In the 1 J/cm^2^ PDT group, the cells showed membrane blebbing (**c**). In the 5 J/cm^2^ PDT group, the cells showed shrinkage (**d**). Cells in the 5 and 15 J/cm^2^ PDT groups displayed nuclear condensation (**i**,**j**).

**Figure 6 cancers-11-00636-f006:**
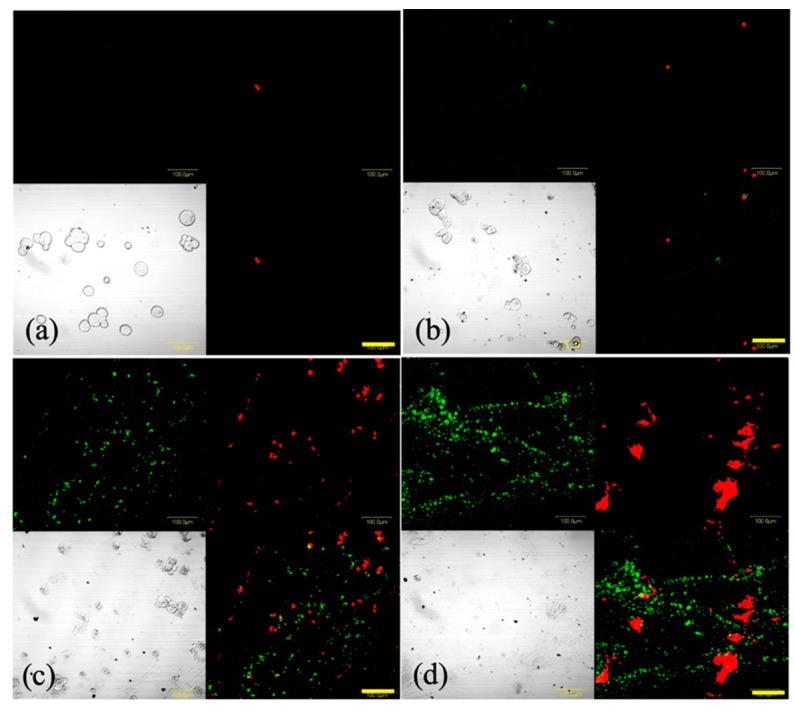
Representative images of SNP cells stained with Annexin V-fluorescein isothiocyanate (green) and ethidium homodimer III (red) following photodynamic therapy (PDT). Cells were incubated with 8 μg/mL glucose-conjugated chlorin e6 (G-Ce6) for 24 h. Following washing with fresh medium, the cells were irradiated with 650-nm laser light (10 mW/cm^2^; 0, 1, 5, or 15 J/cm^2^). Following 4 h of PDT, the cells were stained using the Promokine Apoptotic/Necrotic Cells Detection kit. The images depict (**a**) 8 μg/mL G-Ce6 and 0 J/cm^2^ laser energy, (**b**) 8 μg/mL G-Ce6 and 1 J/cm^2^ laser energy, (**c**) 8 μg/mL G-Ce6 and 5 J/cm^2^ laser energy, and (**d**) 8 μg/mL G-Ce6 and 15 J/cm^2^ laser energy. Scale bar, 100 μm.

**Figure 7 cancers-11-00636-f007:**
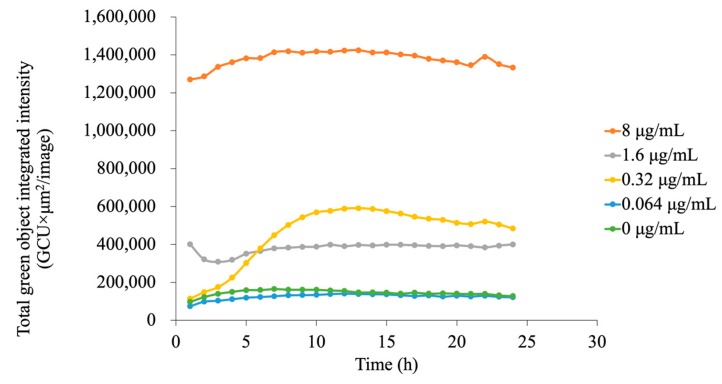
Assessment of apoptosis in the kinetics experiment. Apoptosis induced by glucose-conjugated chlorin e6 (G-Ce6)-mediated photodynamic therapy was determined by an Annexin V-fluorescein isothiocyanate fluorescence study using a live-cell analysis system. The cells were incubated with various concentrations of G-Ce6 (0, 0.064, 0.32, 1.6, or 8.0 μg/mL) for 24 h at 37 °C. After washing with fresh media, the cells were irradiated with 650-nm laser light (10 mW/cm^2^; 5 J/cm^2^). The x-axis represents time (h), and the y-axis represents the total green object integrated intensity. Apoptosis was dependent on the concentration of G-Ce6. GCU, green calibrated unit.

**Figure 8 cancers-11-00636-f008:**
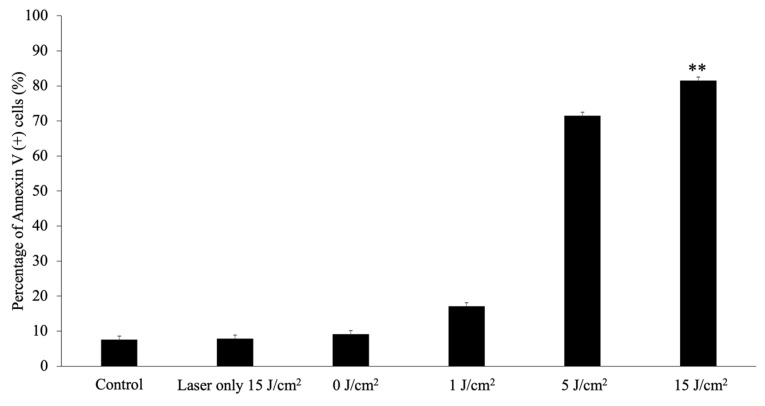
Analysis of apoptosis. Apoptosis was assessed 4 h after laser irradiation using the Muse^®^ Annexin V and Dead Cell Assay kit. The cells were incubated with 8.0 μg/mL glucose-conjugated chlorin e6 for 24 h at 37 °C. After washing with fresh media, the cells were irradiated with 650-nm laser light (10 mW/cm^2^; 0, 1, 5, 15 J/cm^2^). Data were analyzed using Dunn’s multiple comparison test (* *p* < 0.05; control vs. 15 J/cm^2^). The results are presented as the mean ± standard deviation.

**Figure 9 cancers-11-00636-f009:**
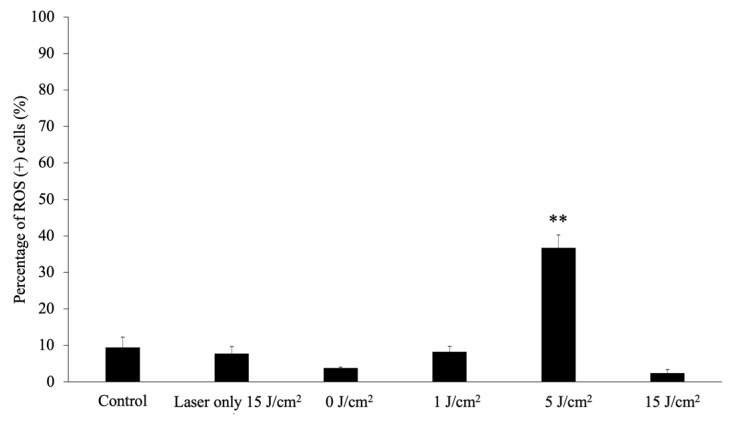
Reactive oxygen species assay. Reactive oxygen species (ROS) generation was assessed 4 h after laser irradiation using the Muse^®^ Oxidative Stress kit. The cells were then incubated with 8.0 μg/mL glucose-conjugated chlorin e6 for 24 h at 37 °C. After washing with fresh media, the cells were irradiated with 650-nm laser light (10 mW/cm^2^; 0, 1, 5, 15 J/cm^2^). Data were analyzed using Dunn’s multiple comparison test. (** *p* < 0.01; 5 J/cm^2^ vs. 15 J/cm^2^). The results are presented as the mean ± standard deviation.

**Figure 10 cancers-11-00636-f010:**
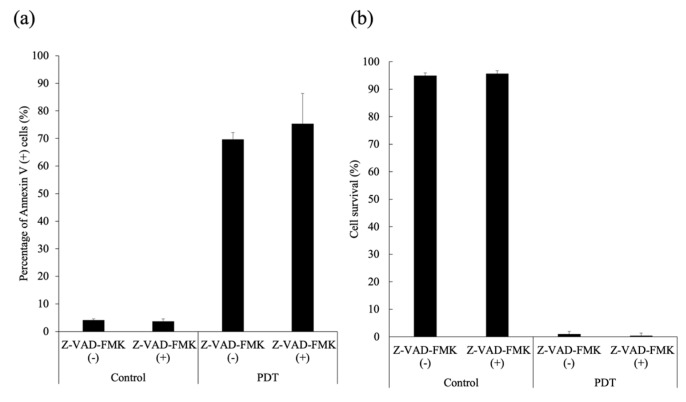
Cytotoxicity in the presence or absence of a pan-caspase inhibitor. SNP cells were incubated with 8 μg/mL glucose-conjugated chlorin e6 for 24 h in the presence (+) or absence (−) of 10 μM Z-VAD-FMK. After rinsing with fresh medium, the cells were irradiated with 650-nm laser light (10 mW/cm^2^; 5 J/cm^2^). (**a**) The percentage of apoptotic cells; and (**b**) cell viability (%). The results are presented as the mean ± standard deviation.

**Figure 11 cancers-11-00636-f011:**
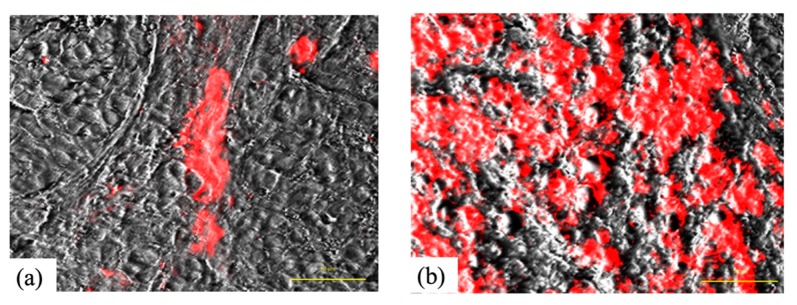
Intratumoral localization of glucose-conjugated chlorin e6 in SNP tumors 5 min and 3 h after administration. (**a**) Glucose-conjugated chlorin e6 (G-Ce6) fluorescence was observed within the tumor blood vessel 5 min after administration. (**b**) G-Ce6 fluorescence was apparent in both tumor interstitial tissue and tumor cells 3 h following administration. Scale bar, 50 μm.

**Figure 12 cancers-11-00636-f012:**
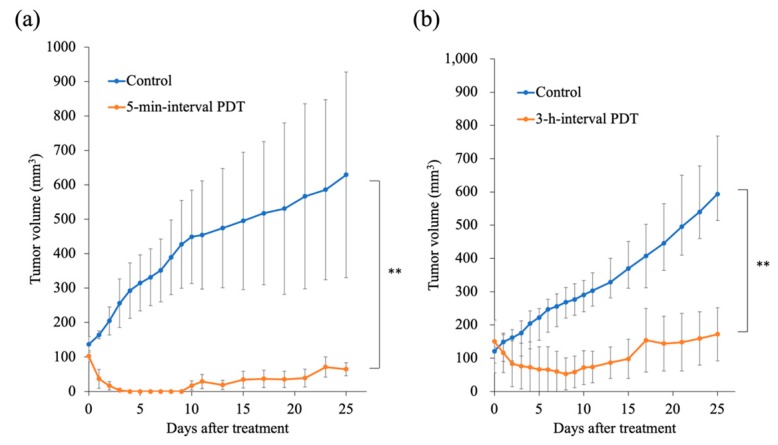
Changes in tumor volume in response to photodynamic therapy (PDT). (**a**) The 5-min-interval PDT (*n* = 4). (**b**) The 3-h-interval PDT (*n* = 3). Data were analyzed using Bonferroni’s multiple comparison test. The results are presented as the mean ± standard deviation. Data were analyzed using Bonferroni’s multiple comparison test (** *p* < 0.01).

**Figure 13 cancers-11-00636-f013:**
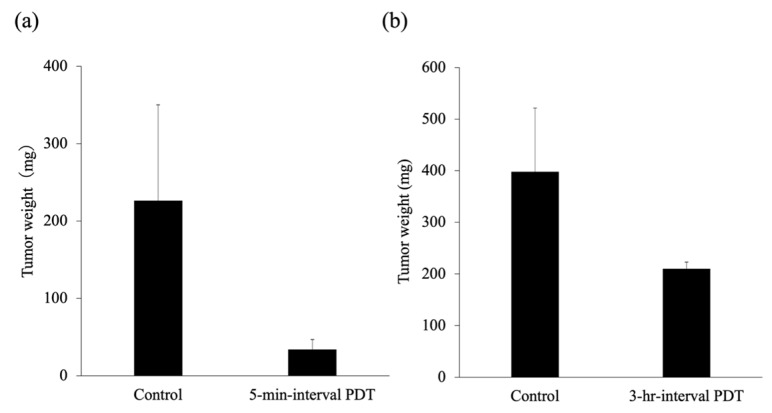
The mean tumor weight 25 days after photodynamic therapy (PDT). (**a**) The 5-min-interval PDT. (**b**) The 3-h-interval PDT. The results are presented as the mean ± standard deviation.

**Figure 14 cancers-11-00636-f014:**
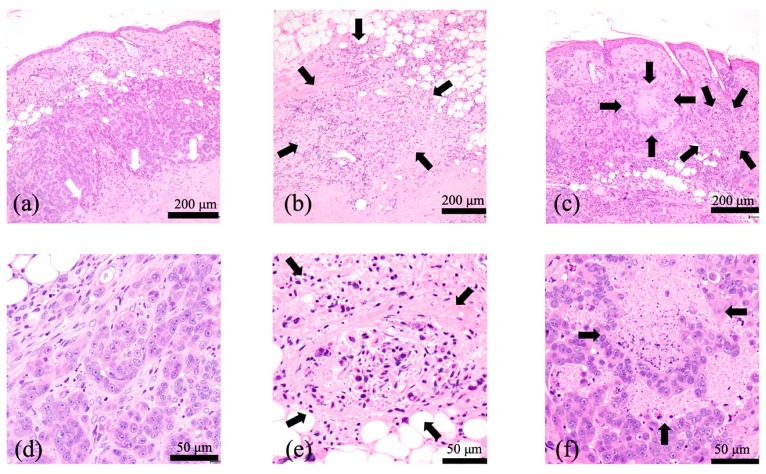
Representative hematoxylin & eosin stained images of the SNP tumors after photodynamic therapy (PDT). (**a**,**d**) Untreated tumors. Tumor cells infiltrated the dermis and necrotic tumor cells occupied the center region of the tumors (white arrows). (**b**,**e**) At 24 h after the 5-min-interval PDT. Widespread tumor cells appeared necrotic (black arrows). (**c**,**f**) At 24 h following the 3-h-interval PDT. Tumor cells at the superficial tumor tissue appeared necrotic (black arrows), whereas deep-seated tumor cells appeared intact. Widespread tumor cells appeared necrotic. (**a**–**c**) Scale bar = 200 μm. (**d**–**f**) Scale bar = 50 μm.

**Figure 15 cancers-11-00636-f015:**
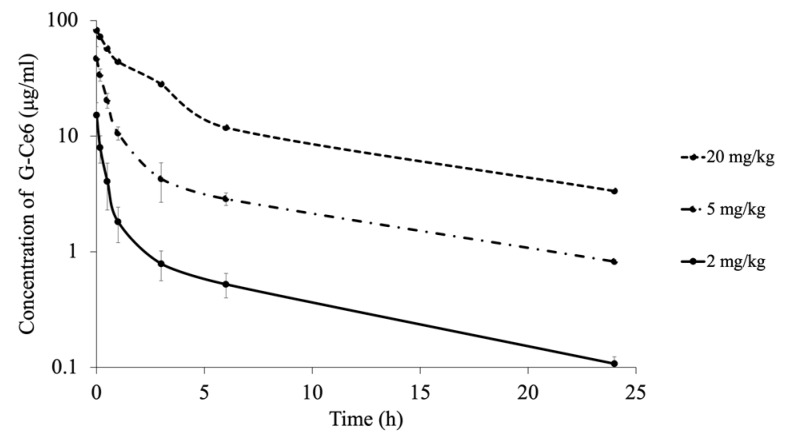
The mean concentrations of glucose-conjugated chlorin e6 (G-Ce6) in the plasma after intravenous administration of three doses of G-Ce6. The volume of G-Ce6 required to achieve the desired doses was diluted with 0.5% saline to obtain a total injection volume of 10 mL. The full injection volume was administered intravenously over 10 min at a rate of 60 mL/h; 2 mg/kg (*n* = 6), 5 mg/kg (*n* = 3), and 20 mg/kg (*n* = 1).

**Table 1 cancers-11-00636-t001:** Pharmacokinetic parameters of dogs after intravenous administration of three doses of glucose-conjugated chlorin e6.

Dose	C_max_ (μg/mL)	T_1/2_ (h)	AUC (μg/mL∙h)	CL (mL/h∙kg)	Vdss (mL/kg)
2 mg/kg	15.19 ± 4.44	3.02 ± 0.58	12.03 ± 3.18	175.90 ± 45.68	529.19 ± 188.67
5 mg/kg	46.99 ± 12.2	2.84 ± 0.60	59.66 ± 7.87	84.75 ± 10.73	258.11 ± 65.57
20 mg/kg	82.05	2.51	234.75	85.20	284.40

C_max_, maximum plasma concentration; T_1/2_, half-life of elimination; AUC, area under the curve; Cl, clearance; Vdss, mean steady state volume of distribution; 2 mg/kg (*n* = 6), 5 mg/kg (*n* = 3), and 20 mg/kg (*n* = 1). The results are presented as the mean ± standard deviation.

**Table 2 cancers-11-00636-t002:** Blood analysis of dogs pre- and post-intravenous administration of three doses of glucose-conjugated chlorin e6.

Parameters	Range	Dose					
		2 mg/kg		5 mg/kg		20 mg/kg	
		Pre	Post	Pre	Post	Pre	Post
Glu (mg/dL)	75–128	99.5 ± 17.85	99.5 ± 17.85	98.0 ± 14.38	96.0 ± 10.88	98.0	107.0
AST (U/L)	17–44	32 ± 4.72	32 ± 4.72	36.5 ± 5.07	31.0 ± 10.47	36.0	55.0
ALT (U/L)	17–78	40 ± 7.17	40 ± 7.17	42.0 ± 11.1	59.0 ± 9.52	41.0	135.0
ALP (U/L)	47–258	154 ± 59.69	154 ± 59.69	136.5 ± 34.97	141.0 ± 47.04	87.0	134.0
GGT (U/L)	5–14	1.5 ± 2.35	1.5 ± 2.35	5.5 ± 1.71	5.0 ± 1.63	8.0	14.0
BUN (mg/dL)	9.2–29.2	15.8 ± 4.20	15.8 ± 4.20	13.2 ± 2.85	13.4 ± 5.05	8.3	8.6
Cre (mg/dL)	0.4–1.4	0.5 ± 0.09	0.5 ± 0.09	0.5 ± 0.05	0.5 ± 0.13	0.5	0.5

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
