# Peer review of "A Basic Study of Photodynamic Therapy with Glucose-Conjugated Chlorin e6 Using Mammary Carcinoma Xenografts"

_cancers, 2019, doi:10.3390/cancers11050636_

Reviewer 1 Report

Dear Authors

The authors developed photodynamic therapy (PDT) effect using glucose-conjugated chlorin e6 (G-Ce6) for large animal model with dog (canine mammary gland tumor). Large animal models have many advantages for useful and important information to reach clinical setup. Also glucose-conjugated chlorins are promising candidate for developing 3rd generation photosensitizers in PDT research. In this manuscript, the authors well organized and described interesting research methods and results. Therefore, I suggest that this paper maybe acceptable for publication in this journal after revision like below.

1. The most important thing in this paper is use of large animal model (dogs). However, in introduction part, there are no related sentences for advantages of using large animal models. Therefore, I suggest that it is need to add related sentences and references for advantages of using large animal models such as, longer life span, physiological parameters are much closer to humans, easy to modeling specific human disease, ... etc. Also are there any large animal models using G-Ce6 from other groups? If there are any related references please add and compare each other. If this is the first paper using large animal model using G-Ce6, please emphasize and mention it.

2. In Discussion part, please add more sentences to explain differences with the ref. 27 to emphasize novelty of this paper.

3. In 4.2. Photosensitizer part, it is need to add related characterization data for G-Ce6 such as, 1H-NMR, UV-Vis, EA or HRFABMS, … etc. Also it should be mentioned to confirm purity of G-Ce6 used in this study.

4. I just wondering water solubility of G-Ce6, because there is no picture of solutions in PBS and/or FBS, and water. Therefore, please add pictures of G-Ce6 solutions in PBS and/or FBS, and water.

5. There is no Conclusion paragraph, therefore it is need to add related paragraph at line 323 (before 4. Materials and Methods).

6. Minor corrections like below.

1)     Please combine from line 69~72 to make one sentence, because the two sentences are the same reference (27).

2)     Figure captions in Figures 1~10, 14~15, all related sentences are placed together. For example, line 77 should be moved to the end of line 76.

3)     Figure captions in Figure 2 and 3, please change like below.

“Figure 2. Subcellular localization of glucose-conjugated chlorin e6 in mitochondria. …”

“Figure 3. Subcellular localization of glucose-conjugated chlorin e6 in lysosome. …”

4)     Line 155, 0.032 mg/mL should be changed to 0.32 mg/mL.

5)     Figures 8 and 9, right of Control, “Laser 15 J/cm2” should be changed to “Laser only 15 J/cm2”.

6)     Figures 12~14, please exchange position of both figures (a) and (b) to make order of 5-min-interval PDT followed by 3-hr-interval PDT.

7)     Figure 12, please add all error bars for negative area as well.

8)     Please add more related references published in 2018.

Thanks so much.

Author Response

Responses to Reviewer comments

Reviewer 1

The authors developed photodynamic therapy (PDT) effect using glucose-conjugated chlorin e6 (G-Ce6) for large animal model with dog (canine mammary gland tumor). Large animal models have many advantages for useful and important information to reach clinical setup. Also glucose-conjugated chlorins are promising candidate for developing 3rd generation photosensitizers in PDT research. In this manuscript, the authors well organized and described interesting research methods and results. Therefore, I suggest that this paper maybe acceptable for publication in this journal after revision like below.

1. The most important thing in this paper is use of large animal model (dogs). However, in introduction part, there are no related sentences for advantages of using large animal models. Therefore, I suggest that it is need to add related sentences and references for advantages of using large animal models such as, longer life span, physiological parameters are much closer to humans, easy to modeling specific human disease, ... etc. Also are there any large animal models using G-Ce6 from other groups?If there are any related references please add and compare each other. If this is the first paper using large animal model using G-Ce6, please emphasize and mention it.

Response: We thank the Reviewer for this specific suggestion. We have added some sentences as follows. 

 “Comparative oncology has been reported to be a quickly expanding filed, which examines both cancer risk and tumor development across species [29]. Canine mammary carcinomas (CMCs) occur spontaneously and have similar clinical presentation and pathophysiology to human cancers [29,30]. In particular, canine invasive mammary carcinomas have a short disease course; therefore, dogs are potent, valuable, and spontaneous organisms for use as caner models to test new therapy [30,31]. Therefore, it was considered that studying dogs with naturally occurring cancer may provide a valuable perspective, which is distinct from that generated when studying human or rodent cancers alone [32].”

2. In Discussion part, please add more sentences to explain differences with the ref. 27 to emphasize novelty of this paper.

 Response: We thank the Reviewer for this specific suggestion. We have added some sentences as follows. 

“Nishie et al. reported the therapeutic effect of PDT with G-Ce6 using a murine model [27]. However, to our knowledge, the present study is the first study using G-Ce6 for CMCs and pharmacokinetics study of G-Ce6 using healthy dogs. PDT using G-Ce6 was effective for CMCs and G-Ce6 was rapidly cleared from the plasma. These results suggested that CMCs is a useful model for preclinical research in comparative oncology and would be an optimal design for human clinical trials.”

3. In 4.2. Photosensitizer part, it is need to add related characterization data for G-Ce6 such as, 1H-NMR, UV-Vis, EA or HRFABMS, … etc. Also it should be mentioned to confirm purity of G-Ce6 used in this study.

 Response: We thank the Reviewer for this specific suggestion. We have added UV-vis. Datum and HPLC datum.

4. I just wondering water solubility of G-Ce6, because there is no picture of solutions in PBS and/or FBS, and water. Therefore, please add pictures of G-Ce6 solutions in PBS and/or FBS, and water.

Response: We thank the Reviewer for this specific suggestion. We have reflected this comment by adding white light image of 2 mg/ml G-Ce6 solution (20% Tween 80 in PBS).

5. There is no Conclusion paragraph, therefore it is need to add related paragraph at line 323 (before 4. Materials and Methods).

 Response: We thank the Reviewer for this specific suggestion. We have added Conclusion paragraph.

6. Minor corrections like below.

1)     Please combine from line 69~72 to make one sentence, because the two sentences are the same reference (27).

Response: We thank the Reviewer for this specific suggestion. In accordance with the reviewer's request, we have had the manuscript rewritten.

“Nishie et al. reported the therapeutic effect of PDT with newly developed glucose-conjugated chlorin e6 (G-Ce6; Figure 1) which had improved water solubility and rapid excretion from the body [27].”

2)     Figure captions in Figures 1~10, 14~15, all related sentences are placed together. For example, line 77 should be moved to the end of line 76.

Response: We thank the Reviewer for this specific suggestion. We have reflected this comment.

3)     Figure captions in Figure 2 and 3, please change like below.

“Figure 2. Subcellular localization of glucose-conjugated chlorin e6 in mitochondria. …”

“Figure 3. Subcellular localization of glucose-conjugated chlorin e6 in lysosome. …”

Response: We thank the Reviewer for this specific suggestion. We have reflected this comment.

4)     Line 155, 0.032 mg/mL should be changed to 0.32 mg/mL.

Response: We thank the Reviewer for this specific suggestion. We have reflected this comment.

5)     Figures 8 and 9, right of Control, “Laser 15 J/cm2” should be changed to “Laser only 15 J/cm2”.

Response: We thank the Reviewer for this specific suggestion. We have reflected this comment.

6)     Figures 12~14, please exchange position of both figures (a) and (b) to make order of 5-min-interval PDT followed by 3-hr-interval PDT.

Response: We thank the Reviewer for this specific suggestion. We have changed position of both figures (a) and (b) to make order of 5-min-interval PDT followed by 3-hr-interval PDT.

7)     Figure 12, please add all error bars for negative area as well.

Response: We thank the Reviewer for this specific suggestion. We have added all error bars for negative area as well.

8)     Please add more related references published in 2018.

Response: We thank the Reviewer for this specific suggestion. We have added some references published in 2018.

Reviewer 2 Report

A basic study of photodynamic therapy with glucose-conjugated chlorin e6 for canine mammary gland tumor.

This article describes the efficacy of PDT using glucose-conjugated chlorin e6 (G-Ce6) in canine cancer line and canine xenografts as well as the pharmacokinetics of PDT-G-Ce6 in canine. This modality has been used in the previous publication to assess the uptake of the drug and the efficacy of this modality on GI cancer models. It is an interesting concept to use a glucose-conjugate to increase its accumulation in the cancer cells and improve the anti-cancer effect, however, the current form of manuscript is not optimal for readers and needs substantial revision. Overall, the quantitative analyses of data are missing therefore not convincing. In addition, PDT resulting in cell apoptosis/necrosis rapidly is not new.

Title: should be stated as mammary carcinoma xenografts or breast tumor xenografts instead of canine mammary gland tumor.

Fig 2&3: need to show more cells in the field and quantify. Not convincing with the current figures.

Fig 5-8, Fig 10: all about the same question if PDT induces apoptosis/necrosis. Put all of these data under one figure,

Fig12: how many mice were used for control group? The number of mice is too small at n=3 or 4. And combine with Fig 13. Fig 12 and 13 are from the same experiment.

Fig 14: need to include an apoptosis marker staining of these tissue samples post treatment like with TUNEL.

Author Response

Responses to Reviewer comments

Reviewer 2

This article describes the efficacy of PDT using glucose-conjugated chlorin e6 (G-Ce6) in canine cancer line and canine xenografts as well as the pharmacokinetics of PDT-G-Ce6 in canine. This modality has been used in the previous publication to assess the uptake of the drug and the efficacy of this modality on GI cancer models. It is an interesting concept to use a glucose-conjugate to increase its accumulation in the cancer cells and improve the anti-cancer effect, however, the current form of manuscript is not optimal for readers and needs substantial revision. Overall, the quantitative analyses of data are missing therefore not convincing. In addition, PDT resulting in cell apoptosis/necrosis rapidly is not new.

Title: should be stated as mammary carcinoma xenografts or breast tumor xenografts instead of canine mammary gland tumor.

Response: We thank the Reviewer for this specific suggestion. Title was stated as mammary carcinoma xenografts instead of canine mammary gland tumor.

We have also replaced the term” canine mammary gland tumor” throughout the paper with “canine mammary carcinoma (CMC)” to use more precise terms.

Fig 2&3: need to show more cells in the field and quantify. Not convincing with the current figures.

Response: We thank the Reviewer for this specific suggestion. We have changed the figures showing more cells. Regrettably, we could not quantify in our fluorescence microscopy.

Fig 5-8, Fig 10: all about the same question if PDT induces apoptosis/necrosis. Put all of these data under one figure,

Response: We thank the Reviewer for this specific suggestion. In our in vitro experimental condition, PDT with G-Ce6 induced mainly necrosis. Therefore, we show these data separately.

Fig12: how many mice were used for control group? The number of mice is too small at n=3 or 4. And combine with Fig 13. Fig 12 and 13 are from the same experiment.

Response: You have raised an important point. We have done experiment at another time and used expensive SCID mice. Therefore, we have done in vivo experiment at minimum number.

Fig 14: need to include an apoptosis marker staining of these tissue samples post treatment like with TUNEL.

Response: In vitro study, PDT with G-Ce6 induced mainly necrosis. in vivo study, 5-min-interval PDT induced necrosis by vascular shutdown. Therefore, we have not stained tumor tissues with TUNEL after PDT.

Round  2

Reviewer 2 Report

It is a descriptive study without proper quantification effort. Specifically, the H&E slides of in vivo study could have been supplemented with any markers to address cell killing, therefore areas of ablation effect.

Author Response

We wish to express our appreciation to the reviewers for their insightful comments on our paper. The comments have helped us significantly improve the paper. 

Responses to Reviewer comments

Reviewer 2

It is a descriptive study without proper quantification effort. Specifically, the H&E slides of in vivo study could have been supplemented with any markers to address cell killing, therefore areas of ablation effect.

Response: We thank the Reviewer for this specific suggestion. We have added some markers in Figure 14.